# Exploring burn first aid knowledge and water lavage practices in Uganda: A cross-sectional study

Brian Kasagga[1,2,3]*, Joseph Baruch Baluku[4], Felix Bongomin[5], Derrick Kasozi[6], Eria Muwanguzi[6], Mercy Namazzi[7], Yusuf Sadiq[2], Rose Alenyo[8], Edris Wamala Kalanzi[8], Darius Balumuka[9], Alex Emmanuel Elobu[1,3]

1 Department of Surgery, Mulago National Referral Hospital, Kampala, Uganda, 2 Women's Hospital International and Fertility Centre, Kampala, Uganda, 3 Academics and Research Department, Society of Uganda Gastrointestinal and Endoscopic Surgeons (SUGES), Kampala, Uganda, 4 Division of Pulmonology, Kiruddu National Referral Hospital, Kampala, Uganda, 5 Faculty of Medicine, Department of Medical Microbiology and Immunology, Gulu University, Gulu, Uganda, 6 Makerere University, School of Medicine, Kampala, Uganda, 7 Kampala International University, College of Health Sciences, Kampala, Uganda, 8 Department of Plastic and Reconstructive Surgery, Kiruddu National Referral Hospital, Kampala, Uganda, 9 Department of Plastic and Reconstructive Surgery, Oregon Health & Science University, Portland, Oregon, United States of America

* briankasagga@gmail.com

**Data Availability Statement:** The data underlying the results presented in the study are available from a Figshare repository at; DOI: 10.6084/m9.

## Abstract

### Background

Low- and middle-income countries experience higher burn-related morbidity and mortality compared to high-income countries. Prehospital Burn First Aid (BFA) improves outcomes. We assessed BFA knowledge and water lavage practices and their associated factors among burn victims, caregivers, and visitors at a tertiary health facility in Uganda.

### Methods

A cross-sectional study was conducted at the Burns Unit of Kiruddu National Referral Hospital in Kampala between 1st April 2022 and 30th November 2022. Participants included burn patients, caregivers, and hospital visitors. Data on BFA knowledge and practices were collected using an interviewer-administered questionnaire. BFA knowledge was evaluated using 13 questions, with a ≥80% score considered adequate. Logistic regression was used to assess for associations.

### Results

We enrolled 404 participants, comprising 68 (16.8%) burn victims, 161(39.9%) primary caregivers, and 175 (43.3%) hospital visitors. Overall, 339 (83.9%) participants had never received BFA information, and 392 (97.0%) had no first aid training. The mean BFA knowledge score was 56±13.9%, with only 5.4% of the participants demonstrating adequate knowledge. Only 26 (27.7%) of current and former burn victims used water lavage as BFA. No statistically significant associations were found between BFA knowledge, water lavage

figshare.27629490 URL: https://figshare.com/s/69df3280c42bfeb0818b.

**Funding:** The author(s) received no specific funding for this work.

**Competing interests:** NO authors have competing interest.

usage, and demographic variables at univariable and multivariable binary logistic regression analyses.

## Conclusion

We highlight inadequate BFA knowledge and practices among victims of burns, their care-givers, and the general population. Addressing these deficiencies through community-based initiatives is crucial to improving burn care in Uganda.

## Introduction

Flames, hot liquids, hot gases, hot surfaces, cold, corrosive chemicals, electricity, lightning, and radiation can all result in burn injuries [1]. The severity of damage caused by burns is determined by the energy of the causative agent and the duration of exposure. The skin is the most commonly affected tissue during burns and accounts for the majority of damage. However, in some cases, electricity can cause more extensive tissue damage than what is visible on the skin. Internal damage to the airways can occur from inhalation of smoke or hot gases. Systemic derangements are frequently observed in cases of larger or deeper burns with a total burn surface area (TBSA) exceeding 30% [1,2].

Burn injuries are common worldwide, with over 180,000 annual deaths. Most fatalities occur in low- and middle-income countries (LMICs), where data on burns is scarce, suggesting the actual number may be higher [3]. In sub-Saharan Africa, scalds are the most common cause of burns, accounting for 59%, while flame burns account for 33% [4]. In Uganda, scalds are most common in male children under the age of five, accounting for approximately 11% of unintentional injuries in this age group [5]. Acid casualties, on the other hand, result from assault and occur in older individuals with a median age of 33 [6].

Morbidity and mortality from burns are particularly higher in LMICs compared to high-income countries(HICs), with some studies reporting that mortality is up to eleven times higher in LMICs, compared to HICs [3,7,8]. This is because HICs have implemented evidence-based remedies such as smoke alarms, water heater temperature control, early response, flame retardant children's sleepwear, and effective burn first aid (BFA) and treatment practices [9]. A global burden of disease analytical study by Yakupu et al. found that the years of life lost to premature mortality (YLL) and years of healthy life lost due to disability (YLD) due to burn injuries were 67% and 33%, respectively [7]. Burn victims often suffer from lasting disabilities, disfigurement, and psychiatric disorders, along with social stigma and rejection. Long-term consequences can include an increased risk of cancer and emotional trauma, especially in children, who may be negatively impacted by alopecia and bone contractures. These physical and psychological effects can cause difficulties in social interaction [4].

Burn injuries impose a significant economic burden on LMICs, with treatment costs higher than those for diseases like tuberculosis and HIV [10,11]. In sub-Saharan Africa alone, burn-related losses exceeded $1 billion in 2019, and South Africa spent $26 million treating burns caused by kerosene and charcoal stoves [12]. However, funding for burns is often inadequate, as illustrated by the Initiative for Social Economic Rights (ISER) report indicating that Uganda's burn budget was among the unfunded priorities for 2020–21. External actors, such as non-governmental organizations and private actors, typically provide most of the funding for burns [13,14]. Burns therefore place an additional burden on LMIC governments and health-care systems, which are already struggling with insufficient funding. The socio-economic

impact of burns is compounded by factors such as lost wages, prolonged care for deformities, and emotional trauma [3].

Therefore, Uganda faces several unique challenges in addressing burns. These challenges include overcrowding in slum settlements, and widespread use of open fires and kerosene stoves predisposing people to burns [15,16]. While burn injuries disproportionately affect children, the increasing number of acid violence cases also highlights gender-specific vulnerabilities [6]. Despite the heavy burden of burn injuries, the healthcare system is often underfunded, with burn care not adequately prioritized in national health budgets. This lack of funding forces burn victims and their families to rely on external support from non-governmental organizations for treatment as described earlier [13].

Burns consume many surgical resources, often requiring repeated surgeries. With the limited surgery resources in sub-Saharan countries, the safe bet is to prevent burns in the first place, improve preparedness, and reduce the extent of injury through proper prehospital care (burn first aid). Prehospital BFA has been shown to reduce morbidity and mortality, as well as associated healthcare costs, by reducing tissue damage, the need for surgery, and the overall outcomes [17–19]. In Uganda, as in most other sub-Saharan African countries, burn victims use alternative practices such as herbs due to a lack of access to medical care [20]. Even so, poor first aid practices, such as animal waste, on the contrary, can lead to a worsening of the burn injury with adverse consequences [20–24]. To the best of our knowledge, studies evaluating BFA knowledge and practices in burn victims and the general Ugandan population are extremely scarce, with only a few retrospective chart-review studies [23,25]. Therefore, we aimed to examine BFA knowledge, BFA practices, and their associated factors. These results will help us to design educational strategies for BFA.

## Methods

### Design

A cross-sectional descriptive study using quantitative techniques was conducted between 1$^{st}$ April 2022 and 30$^{th}$ November 2022 in the Emergency, Inpatient, and Outpatient Departments (EPD, IPD, and OPD) of the Burns Unit, Kiruddu National Referral Hospital (KNRH).

### Study setting

The study was conducted at KNRH, one of Uganda's largest hospitals. It offers burns, plastic surgery, radiology, internal medicine, and palliative care services. It is also a Plastic Surgery Teaching Hospital for the College of Surgeons of East and Central Africa (COSECSA) and the Department of Surgery, Makerere University College of Health Sciences. The hospital is located in Makindye Division, Kampala, approximately 13 km by road southeast of Mulago National Referral Hospital. It is Uganda's highest tertiary hospital in the treatment of burns and has a monthly patient volume of approximately 60 patients.

### Study population and sample size

The sample size was calculated using the formula: n = (z^2 * π * (1 - π)) / p^2, where we assumed a population proportion (π) of 50% for adequate burn first aid knowledge, a 95% confidence interval (z), and a margin of error (p) of 5%. We assumed a population proportion (π) of 50% for the sample size calculation, as this provides the most conservative estimate, ensuring the largest possible sample size and accounting for the greatest variability in the population given no prior studies were available in Uganda. This resulted in a required sample size of 384 participants. To account for a possible 5% non-response rate, we increased the sample to 404

participants, who were recruited through convenience sampling. The study included burn patients, their caretakers, and hospital visitors aged 12 years and above from the emergency and burn wards of KNRH, all of whom were invited to participate voluntarily.

## Data collection, sampling procedures and study measurements

Data was collected using a standardized questionnaire administered by the interviewers through a face-to-face interview. Through consecutive sampling, all ED, OPD, and IPD burn subjects who met the eligibility criteria were recruited into the study after informed consent. Their BFA knowledge was assessed using the questionnaire. The independent variables in this study included demographics (sex, age, education level, etc.), receipt of prior BFA information, prior burn experience, first aid training, and prior experience in providing care for burn victims. The dependent variables were BFA knowledge and BFA practices. The participants' level of knowledge was deemed adequate if they scored 80% or higher on a 13-question questionnaire (S1 Appendix) about appropriate first aid for various burn scenarios, including scalds, electricity, and acid. Each correct answer was worth 1 point and incorrect answers were scored 0. The total score was 13, and a calculated score of 80 percent or higher was considered adequate knowledge.

Appropriate practices were assessed through a question about practices following a burn, with a correct response being water usage and an incorrect response being non-usage of water. The questionnaire can be found in S1 Appendix and was developed based on a literature review and reviewed by plastic surgeons at KNRH who are experts in the field. The data collection tool was pre-tested on ten people.

## Data management and analysis

After data collection, the data was entered into ready-made Google forms. The extracted data were cleaned and sorted using Microsoft Excel. The data were then analyzed using SPSS software for Windows version 26 (IBM, New York, NY). The data were summarized with descriptive statistics; Mean and SD for approximately normally distributed data and median and interquartile ranges for non-normally distributed data. Categorical data were presented as frequency and percentage. We used logistic regression analysis to determine factors associated with low BFA knowledge levels. $P < 0.05$ were considered statistically significant.

We conducted both bivariable and multivariable logistic regression to assess factors associated with BFA knowledge and water lavage practices. For BFA knowledge, all relevant factors, including demographic variables and prior exposure to BFA care or training, were initially included in the bivariable analysis. Variables with low sample sizes—such as caring for a burn victim, receiving burn information, and BFA training—were excluded from the final multivariable model to enhance model robustness.

For the analysis of water lavage practices, all variables were included in the bivariate analysis. In the final multivariable model, we excluded variables with very small counts, specifically "region" and "whether participants received BFA training," to improve model fit by addressing issues with low-frequency data and null values.

## Participant consent and ethics approval

The study was approved by the Mulago Hospital Research and Ethics Committee (REC No; MHREC2226). The study was conducted following the Declaration of Helsinki guidelines. Written informed consent was obtained from participants aged 18 years and above, while assent was obtained from participants below 18 years in addition to parental/guardian consent.

## Results

### Characteristics of study participants

A total of 404 participants responded to the survey, of which 68(16.8%) were current burn victims, 161(39.9%) were current primary caregivers of burn victims, and 175 (43.3%) were visitors. 186 (46%) participants had previously experienced a burn, while 54% of respondents had previously cared for burn victims. The larger percentage of respondents were women (55.9%, n = 226). 46.0%(n = 186) of those surveyed lived in Central Uganda. Regarding educational status, the largest share of respondents had completed high school (39.9%, n = 161). The majority of respondents previously had neither received any BFA information (83.9%, n = 339) nor participated in a first aid training course with burns components (97.0%, n = 392), Table 1.

### Causes of Burns among the participants

Among current burn victims, we found that the most common cause of burns was thermal burns, accounting for 50 cases (73.5%), followed by electrical burns with 9 cases (13.2%), chemical burns with 5 cases (7.4%), and lightning with 1 case (1.5%), Fig 1.

### Burn first aid knowledge

The mean Burn First Aid (BFA) knowledge score was 7.4 out of a total possible score of 13 (SD = 1.8), which corresponds to a percentage score of 56% (SD = 13.9)Only 5.4% (n = 22) of those surveyed scored ≥80% and were considered to have adequate knowledge of burn first aid. Only 31.7% (31.7%) indicated water lavage as a method of BFA and 43.6%(n = 176) answered that they would use antibiotics before coming to the hospital. However, only 10% knew the "drop and roll" technique for clothing fires. Regarding knowledge of using water for acid attacks, only 40 (9.9%) answered correctly, Table 2.

### Factors associated with BFA knowledge

In total, 382 respondents (94.6%) were found to possess inadequate BFA knowledge, Fig 2.

Of those with inadequate knowledge, 370 (95.9%) of them had never received first aid training. However, no particular participant characteristic was associated with inadequate knowledge in both bivariable and multivariable analyses, Table 3.

### Factors associated with water lavage usage following burns among current or former burn victims

Only 26(27.7%) of current and former burn victims used water lavage as BFA, Fig 3. Fig 4 shows other substances used by victims of burns.

Overall, the relationships between demographic variables and water lavage usage did not demonstrate significant associations on either bivariate or multivariate analyses, Table 4.

## Discussion

This study aimed to assess BFA knowledge and water lavage practices and their associated factors. BFA can improve burn outcomes by reducing tissue damage, hospitalization time, and surgical interventions required. General BFA knowledge was very low and only 5.4% of participants had adequate BFA knowledge. This was consistent across all participants, regardless of whether they were currently a burn victim or not, with an average score of 56% on BFA knowledge questions. The percentage of those with low BFA knowledge was higher compared with

**Table 1. Demographics of participants.**

| Characteristics | Frequency | Percentage |
|---|---|---|
| **Age in years** | | |
| 12–18 | 60 | 14.9 |
| 19–35 | 231 | 57.2 |
| 36–59 | 106 | 26.2 |
| Above 60 | 7 | 1.7 |
| **Gender** | | |
| Male | 178 | 44.1 |
| Female | 226 | 55.9 |
| **Region of residence in Uganda** | | |
| Central | 261 | 64.6 |
| Northern | 31 | 7.7 |
| Eastern | 47 | 11.6 |
| Western | 65 | 16.1 |
| **Area of residence** | | |
| Rural | 154 | 38.1 |
| Urban | 250 | 61.9 |
| **Highest education** | | |
| Less than primary | 46 | 11.4 |
| Primary | 143 | 35.4 |
| High school | 161 | 39.9 |
| College | 31 | 7.7 |
| University | 21 | 5.2 |
| Masters | 2 | 0.5 |
| **Current job** | | |
| Private work | 126 | 31.2 |
| Employed | 91 | 22.5 |
| Doesn't work | 187 | 46.3 |
| **Received any BFA information before?** | | |
| Yes | 65 | 16.1 |
| No | 339 | 83.9 |
| **Participated in BFA training before?** | | |
| Yes | 12 | 3.0 |
| No | 392 | 97.0 |
| **Experienced a burn injury before?** | | |
| Yes | 186 | 46.0 |
| No | 218 | 54.0 |
| **Have you ever cared for a person with a burn?** | | |
| Yes | 218 | 54.0 |
| No | 186 | 46.0 |

other studies done in Ethiopia, Indonesia, and India where those with poor knowledge were 66.2, 66, and 60 percent, respectively [26–28]. This figure is higher because burn victims, caretakers, and visitors of burn patients in the hospital setting were all interviewed in this study, which increased the likelihood that people who participated in this study lacked the knowledge to begin with. Furthermore, we utilized a Bloom's cut-off score of greater than 80 percent to assess the adequacy of BFA knowledge, as opposed to the greater than 50 percent in the above studies.

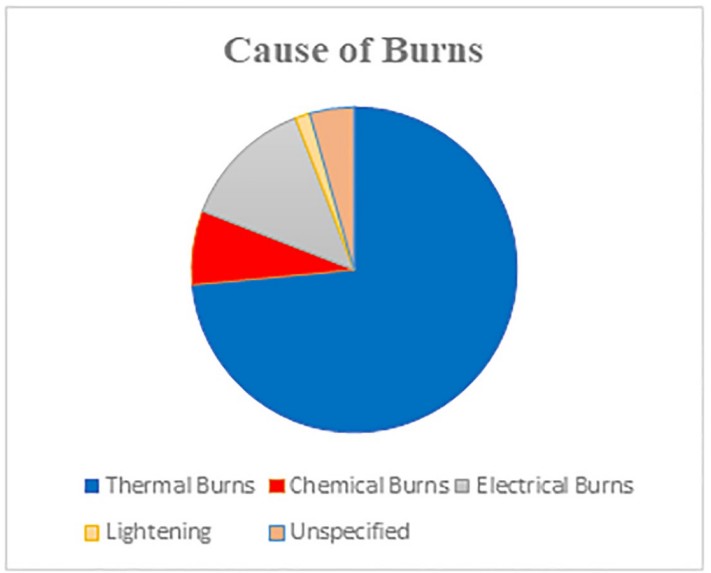

**Fig 1. Aetiology of burns in current burn victims.**

Our study found that an overwhelming majority of participants (over 97%) were aware of the correct first aid steps to take during electric shock incidents, such as avoiding direct contact and promptly turning off the electricity. However, it was surprising to discover that only a small percentage (10%) were familiar with the "drop and roll" technique in case of clothing catching fire. This awareness level was almost significantly lower compared to studies

**Table 2. Percentage of correct answers for the burn knowledge-related questions.**

| Burn knowledge Question | Expected correct answer | Number of correct responses | Percentage |
|---|---|---|---|
| Burns can cause permanent injuries | Agree | 387 | 95.8 |
| Burns can cause mental disorders | Agree | 324 | 80.2 |
| In case of burn injury covering the burned area before heading to hospital can decrease infection risk | Agree | 254 | 62.9 |
| In case of burn injury, picking blisters is an incorrect action | Agree | 224 | 55.4 |
| In case of burn injury Applying first aid at home leads to a better outcome | Agree | 373 | 92.3 |
| It is beneficial to use antibiotics for management | Disagree | 228 | 56.4 |
| In case of electricity burn injury I should not touch the person if he/she is still in electrical current | Agree | 393 | 97.3 |
| In case of electricity burn injury, the first action is to turn off electricity if possible | Agree | 392 | 97.0 |
| 2-year-old boy pulls a kettle of boiling water onto himself | Remove all clothes, and cool with running water for 20 minutes | 97 | 24.0 |
| 10 year old, wanders barefooted into a burning bush as family prepares garden. | Immediately access cool running water and apply for 20 minutes | 95 | 23.5 |
| A 25-year-old man is at a party, he is standing close to a candle and his shirt catches fire | Stop, leave all clothing intact and instruct him to drop to the ground and roll over | 47 | 11.6 |
| 30 year old woman cleaning her swimming pool, acid splashes on her face. What do you do? | Remove clothing and cool in water, shower for 20 min | 40 | 9.9 |
| First aid for a scald | Water lavage | 128 | 31.7 |

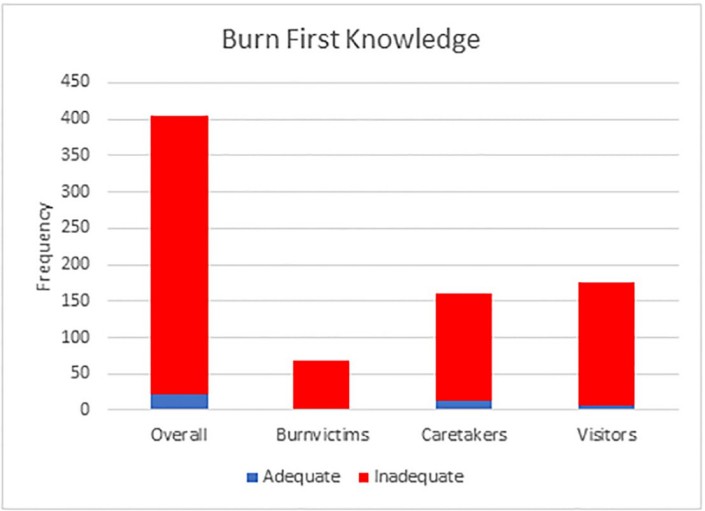

**Fig 2. Assessment of BFA knowledge among burn victims, caregivers, and hospital visitors.**

conducted in Pakistan (20.5%) [29], Bangladesh(80.7) [30] and Vietnam (75%) [31]. This could be probably because of fewer education and safety campaigns regarding fires in Uganda, as indicated by a previous study where they found a majority of schools were unprepared to deal with fires [32].

However, concerning the use of antibiotics, almost half of the respondents answered yes to utilizing antibiotics before seeking medical attention. This pattern resembles the practice of administering antibiotics without proper evaluation from healthcare professionals, which can be counterproductive and contribute to the development of antibiotic resistance [33]. Lastly, the significance of water lavage in acid attack cases was recognized by a mere 10%. These findings shed light on the need for widespread education and awareness campaigns to address gaps in knowledge regarding essential first aid practices, particularly those pertaining to antibiotic use, fire safety, and acid attack response.

According to the published work, the current recommendations for first aid treatment of burn injuries advocate using cold water lavage (between 2 and 15°C) for 15–20 minutes. [19] While over 92% of respondents recognized the benefits of applying first aid at home, the actual practices among burn victims and those with recent burn history were concerning. Only 22.1% of admitted burn patients applied cold water lavage, which is lower compared to similar studies conducted in Ethiopia and Nigeria [26,34]. The use of traditional remedies mirrored patterns observed in other developing countries, but these practices have no supporting evidence for their efficacy [26,34,35]. It is most likely that these substances help to just soothe or numb the pain associated with the burns momentarily.

Furthermore, the utilization of these alternative burn first aid practices complicates wound evaluation and management [36,37]. This is evident in a study conducted in Pakistan, where a substantial number of patients present with severely infected wounds that necessitate excision and skin grafting due to ineffective home treatments [38].

## Strengths and limitations

The study included a limited subgroup of only 68 burn victims, which constrained the exploration of practices specific to this group. We thus added 26 respondents who had experienced burns within the last 12 months. The overall respondent pool consisted of 404 participants,

**Table 3. The association between BFA knowledge and demographic variables.**

| Demographics | Total score Mean(SD) | BFA knowledge, N(%) | | Logistic Regression | | | | |
|---|---|---|---|---|---|---|---|---|
| | | Inadequate | Adequate | Crude Odds Ratio (95% CI) | P-value | Adjusted Odds Ratio (95% CI) | P-value |
| **Overall** | 7.4(1.8) | 382(94.6) | 22(5.4) | NA | - | NA | - |
| **Type of respondent** | | | | | | | |
| Burn victim | 7.0(1.6) | 66(17.3) | 2(9.1) | 1 | | 1 | |
| Care-taker | 7.1(1.5) | 148(38.7) | 13(59.1) | 2.90(0.64–13.21) | 0.169 | 3.25(0.63–16.91) | 0.161 |
| Visitors | 7.1(1.7) | 168(44.0) | 7(31.8) | 1.38(0.278–6.79) | 0.696 | 1.41(0.23–8.05) | 0.696 |
| **Sex** | | | | | | | |
| Female | 7.1(1.6) | 211(51.2) | 15(68.2) | 1.737(0.69–4.36) | 0.25 | 0.57(0.21–1.52) | 0.26 |
| Male | 7.1(1.6) | 171(44.8) | 7(31.8) | 1 | | 1 | |
| **Age** | | | | | | | |
| 12 to 18 | 6.7(1.9) | 57(14.9) | 3(13.6) | 1 | | 1 | |
| 19 to 35 | 7.1(1.6) | 218(51.1) | 13(59.1) | 1.133(0.31–4.11) | 0.85 | 0.79(0.18–3.51) | 0.76 |
| 36 to 59 | 7.2(1.5) | 100(26.2) | 6(27.3) | 1.14(0.275–4.73) | 0.86 | 0.78(0.15–3.95) | 0.76 |
| ≥60 | 6.7(0.8) | 7(1.8) | 0(0) | NA | NA | NA | NA |
| **Region** | | | | | | | |
| Central | 7.3(1.6) | 244(63.9) | 17(77.3) | 1 | | 1 | |
| Northern | 6.6(1.4) | 30(7.9) | 1(4.5) | 0.48(0.06–3.72) | 0.48 | 0.457(0.05–4.17) | 0.49 |
| Eastern | 6.5(1.7) | 44(11.5) | 3(13.6) | 0.98(0.28–3.48) | 0.97 | 1.13(0.24–5.26) | 0.88 |
| Western | 6.7(1.5) | 64(16.8) | 1(4.5) | 0.22(0.029–1.72) | 0.15 | 0.155(0.02–1.35) | 0.09 |
| **Occupation** | | | | | | | |
| Unemployed | 6.7(1.6) | 178(46.6) | 9(40.9) | 1 | | 1 | |
| Private work | 7.1(1.6) | 117(30.6) | 9(40.9) | 1.10(0.33–3.67) | 0.877 | 2.34(0.79–6.9) | 0.125 |
| Employed | 7.3(1.6) | 87(22.8) | 4(18.2) | 1.67(0.50–5.61) | 0.404 | 1.1(0.30–4.07) | 0.88 |
| **Level of education** | | | | | | | |
| Primary | 6.7(1.4) | 177(46.3) | 12(54.5) | 1 | | 1 | |
| Secondary | 7.3(1.7) | 184(48.2) | 8(36.4) | 0.64(0.26–1.61) | 0.34 | 0.61(0.23–1.66) | 0.33 |
| Tertiary | 7.7(1.6) | 21(5.5) | 2(9.1) | 1.4(0.29–6.71) | 0.67 | 5.04(0.72–35.2) | 0.10 |
| **Area of residence** | | | | | | | |
| Rural | 6.8(1.4) | 147(38.5) | 7(31.8) | 1 | | 1 | |
| Urban | 7.3(1.7) | 235(61.5) | 15(68.2) | 1.34(0.53–3.37) | 0.533 | 1.29(0.40–4.13) | 0.67 |
| **Experienced Burn Before** | | | | | | | |
| Yes | 7.2(1.6) | 179(46.9) | 7(31.8) | 0.53(0.21–1.33) | | 0.53(0.21–1.33) | 0.18 |
| No | 6.9(1.6) | 203(53.1) | 15(68.2) | 1 | 0.18 | | |
| **Have you ever received BFA information?** | | | | | | | |
| Yes | 7.7(1.8) | 63(16.5) | 2(9.1) | 0.506(0.12–2.22) | 0.34 | | |
| No | 6.9(1.5) | 319(83.5) | 20(90.9) | | | | |
| **Have you ever received BFA training?** | | | | | | | |
| Yes | 7.6(1.4) | 12(3.1) | 0(0) | NA | NA | NA | NA |
| No | 7.0(1.6) | 370(96.9) | 22(100) | | | | |
| **Have you ever cared for a Burns victim?** | | | | | | | |
| Yes | 7.3(1.5) | 179(46.9) | 7(31.8) | 0.53(0.21–1.33) | | | |
| No | 6.9(1.6) | 203(53.1) | 15(68.2) | 1 | 0.18 | | |

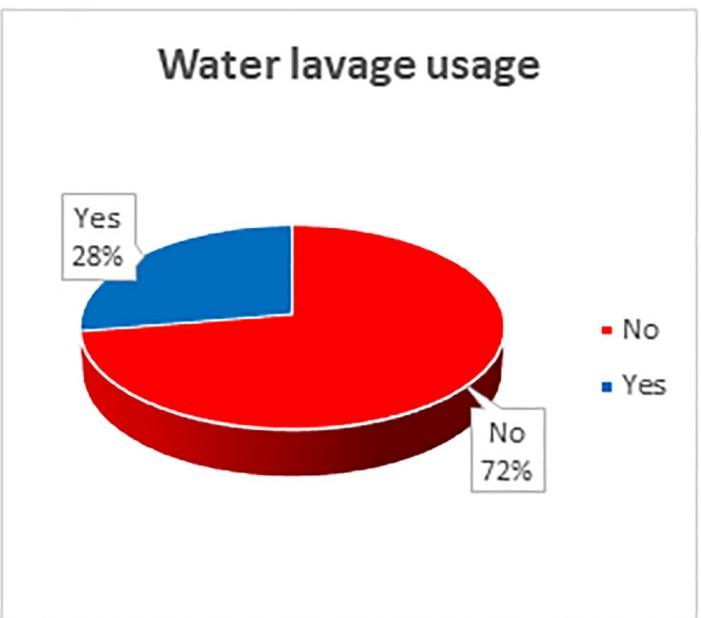

**Fig 3. Proportion of current and former burn victims that used water lavage as the BFA method.**

allowing for an assessment of BFA knowledge among all respondents. It is important to note that this study may be susceptible to recall bias, and this factor should be considered when interpreting the results. We therefore tried to address this by assessing water lavage practices for those who got burned in the last 12 months. While the burn first aid questionnaire was

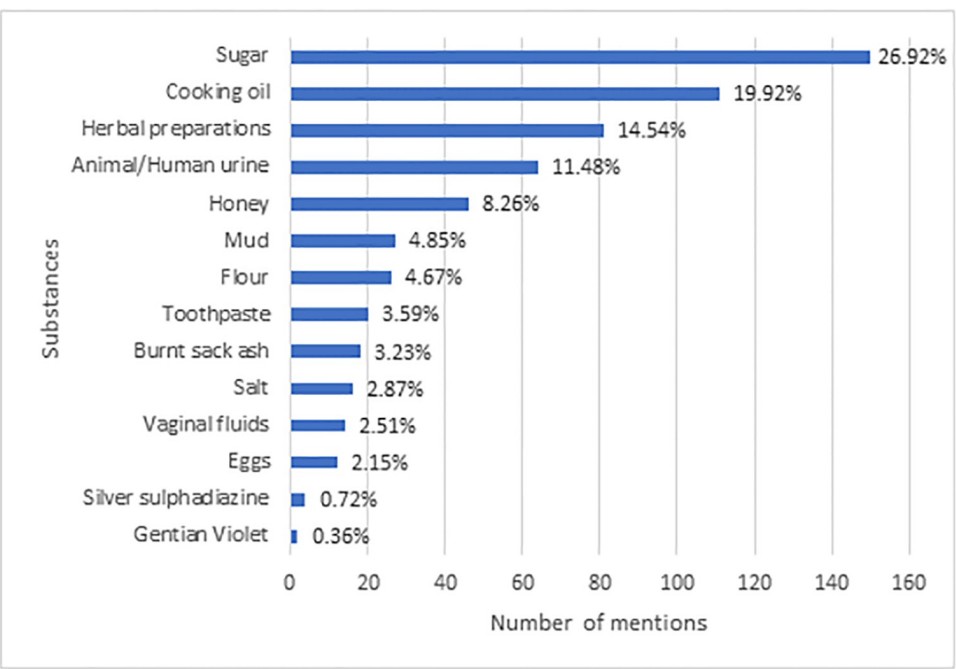

**Fig 4. Frequency of other applied substances following scald burns.**

**Table 4. Factors associated with water lavage usage in current and previous burn patients.**

| Demographics | BFA water lavage practices | | Logistic Regression | | | |
|---|---|---|---|---|---|---|
| | Non-water lavage usage | Water lavage use. | Crude Odds Ratio (95% CI) | P-value | Adjusted Odds Ratio (95% CI) | P-value |
| **Overall** | 68(72.3) | 26(27.7) | NA | | NA | |
| **Sex** | | | | | | |
| Male | 35(51.5) | 11(42.3) | 1 | | | |
| Female | 33(48.5) | 15(57.7) | 1.45(0.58–3.60) | 0.428 | 0.65(0.19–2.16) | 0.483 |
| **Age** | | | | | | |
| 12 to 18 | 17(25.0) | 7(26.9) | 1 | | 1 | |
| 19 to 35 | 34(50.0) | 16(61.5) | 1.14 (0.40–3.30) | 0.805 | 0.90(0.22–3.37) | 0.883 |
| 36 to 59 | 17(25.0) | 3(11.5) | 0.43(0.01–1.94) | 0.272 | 0.10(0.01–1.48) | 0.094 |
| ≥60 | 0(0) | 0(0) | NA | NA | NA | |
| **Region** | | | | | | |
| Central | 46(67.6) | 23(88.5) | 1 | | 1 | |
| Northern | 3(4.4) | 1(3.8) | 0.67(0.07–6.77) | 0.732 | NA | |
| Eastern | 8(11.8) | 0(0) | NA | | NA | |
| Western | 11(16.2) | 2(7.7) | 0.36(0.07–1.78) | 0.212 | 0.18(0.02–1.74) | 0.137 |
| **Occupation** | | | | | | |
| Unemployed | 32(47.1) | 12(46.2) | 1 | | 1 | |
| Private work | 24(35.3) | 4(15.4) | 0.45(0.15–1.31) | 0.143 | 0.35(0.07]-1.79) | 0.207 |
| Employed | 12(17.6) | 10(38.5) | 0.2(0.05–0.77) | 0.200 | 2.69(0.67–10.53) | 0.155 |
| **Level of education** | | | | | | |
| Primary | 38(55.9) | 11(42.3) | 1 | | 1 | |
| Secondary | 28(41.2) | 14(53.8) | 1.73(0.68–4.37) | 0.249 | 1.10(0.34–3.56) | 0.875 |
| Tertiary | 2(2.9) | 1(3.8) | 1.73(0.14–20.89) | 0.667 | 2.05(0.07–63.4) | 0.683 |
| **Area of residence** | | | | | | |
| Rural | 22(32.4) | 6(23.1) | 1 | | 1 | |
| Urban | 46(67.6) | 20(76.9) | 1.59(0.56–4.53) | 0.380 | 0.12(0.01–1.09) | 0.059 |
| **Experienced Burn Before** | | | | | | |
| No | 27(39.7) | 12(46.2) | 1 | | 1 | 0.728 |
| Yes | 41(60.3) | 14(53.8) | 0.77(0.11–1.18) | 0.571 | 0.79(0.21–2.94) | |
| **Have you ever received BFA training?** | | | | | | |
| No | 46(67.6) | 15(57.7) | 1 | | NA | NA |
| Yes | 22(32.4) | 11(42.3) | 8.74(0.87–88.24) | 0.066 | | |
| **Have you ever received any BFA information?** | | | | | | |
| No | 60(88.2) | 21(80.8) | 1 | 0.571 | 1 | 0.819 |
| Yes | 8(11.8) | 5(19.2) | 0.77(0.31–1.91) | | 0.78(0.09–6.85) | |
| **Have you ever cared for a Burns victim?** | | | | | | |
| No | 67(98.5) | 23(88.5) | 1 | | 1 | 0.182 |
| Yes | 1(1.5) | 3(11.5) | 1.53(0.61–3.88) | 0.367 | 2.67(0.63–11.3) | |

developed based on a comprehensive literature review, it was not a validated instrument for assessing BFA. Nevertheless, attempts were made to include representative questions. Lastly, as a hospital-based study, the generalizability of the findings to the broader Ugandan population may be limited although we attempted to include hospital visitors to account for this. Future research should employ stratified sampling or population-based studies across multiple centers to improve the generalizability of the findings.

## Conclusions

This study reveals a concerning lack of BFA knowledge and unsatisfactory practices among burn victims, their caretakers, and visitors in Uganda, highlighting the need for targeted education and awareness campaigns, both in hospitals and within communities. To improve burn care outcomes, we recommend implementing BFA training based on evidence-based BFA practices and education on the dangers of self antibiotic prescription in burn patients. Collaboration between policymakers, healthcare professionals, and community leaders is vital in advocacy and achieving these goals, and promoting better BFA knowledge and practices.

## Supporting information

**S1 Appendix.**
(DOCX)

## Acknowledgments

The authors would like to sincerely thank the burn patients, caregivers, and visitors for their crucial participation in this study. Special appreciation goes to Dr. Anders and Eva Nielson for their invaluable support in enabling the research on burn first aid practices in Uganda. Gratitude is also extended to plastic surgeons at Kiruddu National Referral Hospital for their guidance and critique. The dedicated Kiruddu Plastic Surgery Ward staff's unwavering assistance and support throughout this research are also deeply acknowledged.

## Author Contributions

**Conceptualization:** Brian Kasagga, Yusuf Sadiq, Alex Emmanuel Elobu.

**Data curation:** Brian Kasagga, Mercy Namazzi.

**Formal analysis:** Brian Kasagga.

**Investigation:** Brian Kasagga, Derrick Kasozi, Eria Muwanguzi, Mercy Namazzi, Edris Wamala Kalanzi.

**Methodology:** Brian Kasagga, Joseph Baruch Baluku, Felix Bongomin, Alex Emmanuel Elobu.

**Project administration:** Brian Kasagga, Derrick Kasozi, Eria Muwanguzi.

**Resources:** Brian Kasagga, Joseph Baruch Baluku, Yusuf Sadiq, Edris Wamala Kalanzi.

**Supervision:** Joseph Baruch Baluku, Felix Bongomin, Yusuf Sadiq, Rose Alenyo, Edris Wamala Kalanzi, Darius Balumuka, Alex Emmanuel Elobu.

**Validation:** Brian Kasagga.

**Writing – original draft:** Brian Kasagga, Joseph Baruch Baluku, Derrick Kasozi, Darius Balumuka.

**Writing – review & editing:** Joseph Baruch Baluku, Felix Bongomin, Rose Alenyo, Edris Wamala Kalanzi, Darius Balumuka, Alex Emmanuel Elobu.

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
