## [Decision Letter · Decision Letter 0]

21 Jun 2024

PONE-D-24-18839Exploring Burn First Aid Knowledge and Water Lavage Practices in Uganda: A Cross-Sectional Study.PLOS ONE

Dear Dr. Kasagga,

Thank you for submitting your manuscript to PLOS ONE. After careful consideration, we feel that it has merit but does not fully meet PLOS ONE’s publication criteria as it currently stands. Therefore, we invite you to submit a revised version of the manuscript that addresses the points raised during the review process.

Please address and improve your manuscript by responding to each of the comments from reviewers 1, 2, and 3.

We look forward to receiving your revised manuscript.

Kind regards,

Xiaosheng Tan

Academic Editor

PLOS ONE

Additional Editor Comments:

Dear Authors,

Thank you for choosing PLOS ONE. Please address and improve your manuscript by responding to each of the comments from reviewers 1, 2, and 3.

Sincerely,

Xiaosheng Tan

Reviewers' comments:

Reviewer's Responses to Questions

**Comments to the Author**

1. Is the manuscript technically sound, and do the data support the conclusions?

Reviewer #1: Yes

Reviewer #2: Yes

Reviewer #3: Partly

Reviewer #4: Partly

2. Has the statistical analysis been performed appropriately and rigorously? 

Reviewer #1: No

Reviewer #2: Yes

Reviewer #3: Yes

Reviewer #4: No

3. Have the authors made all data underlying the findings in their manuscript fully available?

Reviewer #1: Yes

Reviewer #2: Yes

Reviewer #3: Yes

Reviewer #4: No

4. Is the manuscript presented in an intelligible fashion and written in standard English?

Reviewer #1: Yes

Reviewer #2: Yes

Reviewer #3: Yes

Reviewer #4: Yes

5. Review Comments to the Author

Reviewer #1: This study highlights inadequate Burn First-aid Knowledge and water lavage practices in Uganda.

1.Is there any periodic temporal trend in the number of burn first aid cases in the hospital?

2.Please make a necessary explanation on the formula for sample size calculation in your paper.

3.Please provide information about the main causes of burn injuries in your dataset.

4.In this research, the logistic regression model was performed on a imbalanced data with 68 urn victims(16.8%). Did you consider using a weighted logistic regression model to address the issue of imbalanced data?

Reviewer #2: This manuscript presents a cross-sectional analysis of prehospital burn first aid (BFA) knowledge among a cohort of 404 participants. It examines the correlation between BFA knowledge and demographic variables, including age, gender, education levels, and geographical region, etc. The authors conducted descriptive analyses and performed logistic regression using the collected data with the conclusion that demographic factors do not exhibit a statistically significant association with BFA knowledge or the application of water lavage. The authors acknowledge a potential sampling bias as the study limitation. The findings underscore an urgent need for enhanced public education and awareness campaigns of essential first aid knowledge, especially pertaining to the appropriate antibiotics use, "drop and roll" technique, and acid attack response. The paper discusses a meaningful topic with data back-up and is written with good structure. I recommend the publication with revisions to the following comments.

1. On page 1 of the manuscript, the results part in the abstract stated that the "Mean BFA knowledge score was 56% (SD 13.9)." Given that the total possible score for the questionnaire is 13, the reported SD of 13.9 is surprisingly high. Should this value be expressed as a percentage (i.e., 13.9%) to reflect the variability relative to the mean percentage score? The same statement was also found on page 5 right under Table 1.

2. On page 2 of the manuscript, in the third paragraph of the introduction, the authors discussed that LMICs experience higher morbidity and mortality from burn injuries compared to HICs. While this assertion underscores the critical need for disseminating BFA knowledge in LMICs, it appears to lack data backing. Could the authors provide data to substantiate this claim? The inclusion of such data comparison would greatly reinforce the argument for the importance of BFA education in LMICs.

3. On page 3 of the manuscript, in the "Study population and sample size" subsection of the Methods. The authors use several mathematical symbols whose meanings may not be immediately clear to all readers.To prevent any potential misinterpretation, it would be beneficial for the authors to provide explicit definitions for these symbols. For instance, could the authors clarify that z represents the Z-score corresponding to a 95% confidence interval, and that π denotes the assumed population proportion? Elaborating on these symbols will enhance the manuscript's clarity and accessibility.

4. In the sample size calculation detailed under the "Study population and sample size" section, the authors have opted to use a π=50% for the potential largest sample size required. Given that the actual population proportion in this study is significantly lower than 50%, a brief explanation of using π=50% would be beneficial for understanding of broader readers.

5. On page 7 of the manuscript, under Figure 1, the authors note that among those with inadequate BFA knowledge, a substantial proportion were from the central region (77.3%, n=17), female (68.2%, n=15), and aged between 19 to 35 years (59.1%, n=13). The statement might be misleading as these demographics (central region, female, and 19-35 years) appear to represent the majority of the study's sample population. Could the authors clarify whether the high percentages reported are indications of these groups' lack of first aid training rather than a reflective of the sample's composition?

6. Some minor comments are as follows.

a). Figure 1 and 2 currently missing captions and labels. Could the authors provide captions and labels to figures to enhance the reader's understanding of these visual elements?

b). In Table 2, row 5, there appears to be a typo: "pricking blisters" instead of "picking blisters."

c). In Table 2, row 8 and row 9, it should be "In case of" instead of "Incase of".

d). On page 3 of the manuscript, within the Methods section under "Data collection, sampling procedures and study measurements" subsection, the fourth row from the bottom, the authors mention several demographic factors (sex, age, education level, level of education, etc.). The terms "education level" and "level of education" are used redundantly. Could the authors consolidate these terms to eliminate repetition?

Reviewer #3: The study conducted in Uganda reveals a significant lack of burn first aid (BFA) knowledge and practice among the surveyed population, highlighting the urgent need for public education and improved access to accurate first aid information. However, there are several questions that need to be answered before considering the publication.

1) The introduction provides sufficient background information, but further discussion of the unique burn issues in Uganda could be added.

2) The study design is clearly described, but the representativeness of the sample should be ensured in order to generalize to a wider Ugandan population.

3) The statistical methods used are appropriate, but a detailed explanation of model selection and assumptions is needed.

4) The discussion should elaborate on the implications of the results and compare them with existing literature.

5) The manuscript should be carefully checked for grammatical errors, spelling mistakes or unclear presentation.

Reviewer #4: In this study, the authors assessed knowledge and practices of Prehospital Burn First Aid (BFA) among 404 participants, including burn victims, caregivers, and visitors. Findings revealed significant deficiencies: 83.9% had never received BFA information, 97.0% lacked first aid training, and only 5.4% demonstrated adequate BFA knowledge. Water lavage, a crucial BFA component, was used by only 27.7% of burn victims. No demographic factors showed significant associations with BFA knowledge or water lavage usage. This study indicated that addressing these gaps through community-based initiatives is crucial for improving burn care in Uganda.

In this study, the authors conducted a statistical description highlighting deficiencies in BFA knowledge and training. However, the study does not include further analysis to offer new insights into mitigating the risk of burns, indicating a lack of novel findings.

6. PLOS authors have the option to publish the peer review history of their article (what does this mean?). If published, this will include your full peer review and any attached files.

Reviewer #1: No

Reviewer #2: No

Reviewer #3: **Yes: **Xiaodong Zou

Reviewer #4: No

---

## [Author Response · Author response to Decision Letter 0]

8 Nov 2024

Dear Xiaosheng Tan

Thank you for reviewing this manuscript.

Thank you for your feedback and guidance. I have addressed the additional requirements as follows:

PLOS ONE Style Requirements: I have carefully revised the manuscript to align with the PLOS ONE style guidelines, ensuring appropriate formatting of the title, authors, affiliations, and other elements. The file names have also been updated as per the guidelines outlined in the PLOS ONE 

Data Availability

I have attached all the data for this manuscript in the Cleaned data file.

Dear Reviewers,

Thank you for the review of this paper. We appreciate the detailed feedback provided by the reviewers. We have carefully addressed each point raised, and the manuscript has been revised accordingly. Below is our point-by-point response:

Reviewer #1:

1. Is there any periodic temporal trend in the number of burn first aid cases in the hospital?

Response: Thank you for this insightful question. However, assessing temporal trends in burn first aid cases was not one of our study objectives. We interviewed a variety of participants, including current burn victims, caretakers, and visitors. Among the burn victims, some had sustained their injuries much earlier and had been on the hospital ward for an extended period, while others had only recently presented with burns. As a result, we did not collect specific data on when the burn injuries occurred, which would have been necessary to evaluate temporal trends.

2. Please make a necessary explanation on the formula for sample size calculation in your paper

Response: The formula used for sample size calculation has now been clarified in the "Study population and sample size" section. We included explanations of the symbols and assumptions (e.g., Z-score, π) as suggested.

The sample size was calculated using the formula: n = (z^2 * π * (1 - π)) / p^2, where we assumed a population proportion (π) of 50% for adequate burn first aid knowledge, a 95% confidence interval (z), and a margin of error (p) of 5%. This resulted in a required sample size of 384 participants. To account for a possible 5% non-response rate, we increased the sample to 404 participants, who were recruited through convenience sampling. The study included burn patients, their caretakers, and hospital visitors aged 12 years and above from the emergency and burn wards of KNRH, all of whom were invited to participate voluntarily. 

In our study, we calculated the sample size using the following formula for estimating proportions:

n = (z² * π * (1-π)) / p², where:

n is the required sample size,

z is the z-score corresponding to the desired confidence level (for a 95% confidence interval, z = 1.96),

π represents the estimated proportion of the population that has adequate knowledge of burn first aid (BFA), which we assumed to be 50% as a conservative estimate when no prior data is available,

p is the margin of error, set at 5% (0.05).

This calculation yielded a minimum sample size of 384 participants. To account for potential non-response or incomplete data (estimated at 5%), we increased the sample size to 404 participants.

We used convenience sampling to enroll all burn patients, their caretakers, and visitors aged 12 years and above present in the emergency and burn wards of KNRH. While convenience sampling was practical for this setting, we acknowledge that it may limit the generalizability of the results, and this is addressed in the study’s limitations.

Line 128-144

3. Provide information on the main causes of burn injuries in the dataset.

Response: We have added a new subsection in the results addressing the common causes of burn injuries, as derived from our dataset with a table

We sought to determine what current burn victims cause of burns. We also added a graph to summarise the cause as advised. Thank you for the suggestion again.

Line 203-208

4. In this research, the logistic regression model was performed on a imbalanced data with 68 burn victims(16.8%). Did you consider using a weighted logistic regression model to address the issue of imbalanced data?

Response Thank you for your valuable insight regarding the imbalanced data in our study and the need for a weighte logistic regression model

Our population comprised current burn victims, caretakers, and visitors. For the multivariate logistic regression regarding factors associated with burn first aid knowledge, we considered all 404 participants. However, for the multivariate logistic regression on water lavage usage, we only included current burn victims and those who had experienced burns within the past year, resulting in a total of 94 participants in the model. This decision was made because many caretakers and visitors had never experienced burns themselves, making them unsuitable for this particular analysis. We thus constructed the second model to determine factors associated with water lavage usage using 94 participants.

Reviewer #2:

1. On page 1 of the manuscript, the results part in the abstract stated that the "Mean BFA knowledge score was 56% (SD 13.9)." Given that the total possible score for the questionnaire is 13, the reported SD of 13.9 is surprisingly high. Should this value be expressed as a percentage (i.e., 13.9%) to reflect the variability relative to the mean percentage score? The same statement was also found on page 5 right under Table 1.

Response: Thank you for your insightful comment. The reported standard deviation of 13.9 is indeed related to the mean percentage score of 56%. We initially considered presenting the scores as percentages because we defined adequate knowledge as 80%, based on Bloom's cutoff score. However, we recognize that this could lead to confusion for readers. To clarify, we have now included both the actual score and its standard deviation, alongside the percentage score. This should enhance the understanding of the variability in relation to the mean percentage score.

Line 39 and Line 211-213

2. On page 2 of the manuscript, in the third paragraph of the introduction, the authors discussed that LMICs experience higher morbidity and mortality from burn injuries compared to HICs. While this assertion underscores the critical need for disseminating BFA knowledge in LMICs, it appears to lack data backing. Could the authors provide data to substantiate this claim? The inclusion of such data comparison would greatly reinforce the argument for the importance of BFA education in LMICs.

Response: Thank you so much for this correction. We have appropriately added more data to support this claim and references from relevant studies to substantiate the higher morbidity and mortality rates from burn injuries in LMICs compared to HICs.

Line 67-68

3. On page 3 of the manuscript, in the "Study population and sample size" subsection of the Methods. The authors use several mathematical symbols whose meanings may not be immediately clear to all readers.To prevent any potential misinterpretation, it would be beneficial for the authors to provide explicit definitions for these symbols. For instance, could the authors clarify that z represents the Z-score corresponding to a 95% confidence interval, and that π denotes the assumed population proportion? Elaborating on these symbols will enhance the manuscript's clarity and accessibility.

Response: Thank you so much for this suggestion; definitions for z (Z-score) and π (population proportion) have been included to make this section clearer to readers.

We edited the Study population and sample size appropriately to this;

The sample size was calculated using the formula: n = (z^2 * π * (1 - π)) / p^2, where we assumed a population proportion (π) of 50% for adequate burn first aid knowledge, a 95% confidence interval (z), and a margin of error (p) of 5%. This resulted in a required sample size of 384 participants. To account for a possible 5% non-response rate, we increased the sample to 404 participants, who were recruited through convenience sampling. The study included burn patients, their caretakers, and hospital visitors aged 12 years and above from the emergency and burn wards of KNRH, all of whom were invited to participate voluntarily Line 128-144

4. In the sample size calculation detailed under the "Study population and sample size" section, the authors have opted to use a π=50% for the potential largest sample size required. Given that the actual population proportion in this study is significantly lower than 50%, a brief explanation of using π=50% would be beneficial for understanding of broader readers.

Response: Thank you for raising this important point. We opted to use π=50% in our sample size calculation as it provides the most conservative estimate, ensuring the largest possible sample size. This is a standard approach when there is limited prior knowledge about the population proportion, as it maximizes the sample size needed to detect significant differences. Although the actual population proportion in our study turned out to be lower than 50%, using π=50% allowed us to ensure adequate statistical power and robustness of the study's findings. We have added this explanation to the manuscript for clarity.

Line 132-134

5. On page 7 of the manuscript, under Figure 1, the authors note that among those with inadequate BFA knowledge, a substantial proportion were from the central region (77.3%, n=17), female (68.2%, n=15), and aged between 19 to 35 years (59.1%, n=13). The statement might be misleading as these demographics (central region, female, and 19-35 years) appear to represent the majority of the study's sample population. Could the authors clarify whether the high percentages reported are indications of these groups' lack of first aid training rather than a reflective of the sample's composition?

Response: I have edited this section such that we only mention that there were no significant differences in the characteristics.

Line 235-241

6. Some minor comments are as follows.

a). Figure 1 and 2 currently missing captions and labels. Could the authors provide captions and labels to figures to enhance the reader's understanding of these visual elements?

b). In Table 2, row 5, there appears to be a typo: "pricking blisters" instead of "picking blisters."

c). In Table 2, row 8 and row 9, it should be "In case of" instead of "Incase of".

d). On page 3 of the manuscript, within the Methods section under "Data collection, sampling procedures and study measurements" subsection, the fourth row from the bottom, the authors mention several demographic factors (sex, age, education level, level of education, etc.). The terms "education level" and "level of education" are used redundantly. Could the authors consolidate these terms to eliminate repetition?

Response: Captions and labels have been added to Figures 1 and 2. The typos identified in Table 2 and the redundancy in the demographic factors section have been corrected.

Thank you for this correction

Line 150, 223

Reviewer #3: 

The study conducted in Uganda reveals a significant lack of burn first aid (BFA) knowledge and practice among the surveyed population, highlighting the urgent need for public education and improved access to accurate first aid information. However, there are several questions that need to be answered before considering the publication.

1) The introduction provides sufficient background information, but further discussion of the unique burn issues in Uganda could be added.

Response

Response: Thank you so much for this suggestion. We have edited the discussion which includes an overview of the specific challenges related to burn injuries in Uganda.

Line 84-98

2) The study design is clearly described, but the representativeness of the sample should be ensured in order to generalize to a wider Ugandan population.

Response:

Thank you for your valuable feedback regarding the representativeness of our sample. We recognize that there are limitations to generalizing our findings to the broader Ugandan population. Given our data collection was conducted over a period of five months, we aimed to capture a diverse range of participants, including burn victims, caretakers, and hospital visitors. The inclusion of visitors likely reflects the demographic diversity of the Ugandan population. Given that our center is the top referral unit, patients come from various regions of Uganda, as many individuals seek care at our tertiary center, which serves as a referral unit for burn cases across the country.

However, we understand that our sample may not fully represent all segments of the population affected by burns. This limitation has been highlighted in the manuscript's discussion section.

To further improve representativeness, we suggest that future studies consider employing stratified sampling methods or including additional centers to capture a broader demographic spectrum. We appreciate your insightful comment, which has prompted us to clarify these considerations in our revised manuscript.

Line 323-327

3) The statistical methods used are appropriate, but a detailed explanation of model selection and assumptions is needed.

Reply;

For BFA knowledge, we applied both simple bivariable and multivariable logistic regression. Initially, all factors—including demographic variables and prior exposure to BFA care or training—were included in the bivariable analysis. However, variables with small sample sizes, such as caring for a burn victim, receiving burn information, and BFA training, were excluded from the multivariable model to maintain statistical power and model stability.

For the analysis of water lavage practices, we similarly included all variables in the initial bivariate analysis. In the final multivariable model, we excluded variables with very small counts, such as "region" and "whether participants received BFA training," to improve model fit by removing data with low frequency and null values

This has been capture on Lines 172 to 181

4) The discussion should elaborate on the implications of the results and compare them with existing literature.

Reply;

Thank you for this valuable feedback. We have carefully elaborated on the implications of our findings in the discussion. We have also made thorough comparisons with existing literature, emphasizing how our results align or differ from previous studies. Additionally, we have highlighted the need for targeted burn first aid (BFA) education campaigns both in hospital settings and broader communities, as these could play a crucial role in improving BFA knowledge and practices. Line 271-342

5) The manuscript should be carefully checked for grammatical errors, spelling mistakes or unclear presentation.

Response

Thank you for your valuable feedback regarding the clarity and presentation of our manuscript. We acknowledge the importance of ensuring that our work is free from grammatical errors and spelling mistakes, as these can affect readability and comprehension.

In response to your comment, we have conducted a thorough review of the manuscript to identify and correct any such issues. We also sought assistance from other editors to enhance the overall clarity and coherence of the text. We believe these revisions will improve the quality of our presentation and make our findings more clear to the future readers.

Reviewer #4: 

In this study, the authors assessed knowledge and practices of Prehospital Burn First Aid (BFA) among 404 participants, including burn victims, caregivers, and visitors. Findings revealed significant deficiencies: 83.9% had never received BFA information, 97.0% lacked first aid training, and only 5.4% demonstrated adequate BFA knowledge. Water lavage, a crucial BFA component, was used by only 27.7% of burn victims. No demographic factors showed significant associations with BFA knowledge or water lavage usage. This study indicated that addressing these gaps through community-based initiatives is crucial for improving burn care in Uganda.

In this study, the authors conducted a statistical description highlighting deficiencies in BFA knowledge and training. However, the study does not include further analysis to offer new insights into mitigating the risk of burns, indicating a lack of novel findings.

Reply

---

## [Decision Letter · Decision Letter 1]

22 Nov 2024

PONE-D-24-18839R1Burn First Aid Knowledge And Water Lavage Practices Among Victims of Burn Injuries, Their Caregivers, and Visitors at a Tertiary Hospital in Uganda.PLOS ONE

Dear Dr. Kasagga,

Thank you for submitting your manuscript to PLOS ONE. After careful consideration, we feel that it has merit but does not fully meet PLOS ONE’s publication criteria as it currently stands. Therefore, we invite you to submit a revised version of the manuscript that addresses the points raised during the review process.

 Please respond to reviewer 1's comments.

We look forward to receiving your revised manuscript.

Kind regards,

Xiaosheng Tan

Academic Editor

PLOS ONE

Journal Requirements:

Reviewers' comments:

Reviewer's Responses to Questions

**Comments to the Author**

1. If the authors have adequately addressed your comments raised in a previous round of review and you feel that this manuscript is now acceptable for publication, you may indicate that here to bypass the “Comments to the Author” section, enter your conflict of interest statement in the “Confidential to Editor” section, and submit your "Accept" recommendation.

Reviewer #1: All comments have been addressed

Reviewer #2: All comments have been addressed

Reviewer #4: All comments have been addressed

2. Is the manuscript technically sound, and do the data support the conclusions?

Reviewer #1: Partly

Reviewer #2: Yes

Reviewer #4: No

3. Has the statistical analysis been performed appropriately and rigorously? 

Reviewer #1: Yes

Reviewer #2: Yes

Reviewer #4: No

4. Have the authors made all data underlying the findings in their manuscript fully available?

Reviewer #1: Yes

Reviewer #2: Yes

Reviewer #4: Yes

5. Is the manuscript presented in an intelligible fashion and written in standard English?

Reviewer #1: Yes

Reviewer #2: Yes

Reviewer #4: Yes

6. Review Comments to the Author

Reviewer #1: Thanks for the improvement on the paper manuscript and response to interview comments. I have two minor commnets on current version.

1.Please improve the quality of the paper writing.

2.This is a cross-sectional study, why don't you use the title "Exploring Burn First Aid Knowledge and Water Lavage Practices in Uganda: A Cross-Sectional Study"?

10.1101/2023.08.10.23293067

Reviewer #2: (No Response)

Reviewer #4: The authors have addressed the comments from the previous review. However, I cannot recommend this manuscript for publication in PLOS ONE for the following reasons:

The topic addressed in the manuscript does not make a substantial contribution to the field. The study relies on a questionnaire rather than experimental or other quantitative methods, which limits the depth and robustness of the findings. Furthermore, the research question lacks novelty and practical relevance, reducing the overall impact of the study.

The sample size used in the study is insufficient to support robust statistical analysis or draw reliable conclusions. This limitation raises significant concerns about the validity and generalizability of the findings, thereby undermining the scientific rigor of the manuscript.

7. PLOS authors have the option to publish the peer review history of their article (what does this mean?). If published, this will include your full peer review and any attached files.

Reviewer #1: No

Reviewer #2: No

Reviewer #4: No

---

## [Author Response · Author response to Decision Letter 1]

6 Jan 2025

Response to Reviewers

Dear Dr. Xiaosheng Tan and Reviewers,

Thank you so much for the opportunity to revise and resubmit our manuscript. We appreciate the feedback provided by the reviewers and have carefully addressed each comment to improve the manuscript's clarity, rigor, and impact. Below, we provide detailed responses to each point raised by the reviewers, along with a summary of the revisions made to the manuscript.

Reviewer #1

Comment 1: Please improve the quality of the paper writing.

Response: We have carefully revised the manuscript to enhance its readability and ensure that the language is clear, concise, and professional. Specifically, we focused on removing typing errors, improving sentence structure, eliminating redundant phrases, and ensuring consistency in terminology.

Comment 2: This is a cross-sectional study. Why don't you use the title "Exploring Burn First Aid Knowledge and Water Lavage Practices in Uganda: A Cross-Sectional Study"?

Response: We agree with the suggestion and have revised the title to:

"Exploring Burn First Aid Knowledge and Water Lavage Practices in Uganda: A Cross-Sectional Study."

Reviewer #4

Comment 1: The topic addressed in the manuscript does not make a substantial contribution to the field. The study relies on a questionnaire rather than experimental or other quantitative methods, which limits the depth and robustness of the findings.

Response: 

Thank you for this critique. While the study uses a questionnaire-based design, it addresses a critical gap in understanding burn first aid knowledge and practices in Uganda, a region with limited data on community and caregiver knowledge and practices related to burn first aid. This information is essential for informing public health interventions and policies aimed at improving burn outcomes. Moreover, the findings from this study have directly contributed to a Quality Improvement (QI) project at the hospital, where we are working to distribute educational materials, such as posters and booklets, to burn victims and their families. These materials will focus on the correct water lavage practices following burns. Additionally, we plan to initiate community outreach programs and radio sensitizations to further raise awareness about burn prevention and burn furst aid. We have emphasized the significance and practical relevance of this work in the manuscript.

Comment 2: The sample size used in the study is insufficient to support robust statistical analysis or draw reliable conclusions.

Response: 

We acknowledge the concerns regarding the sample size.. We have clarified the sample size calculation and its justification in the Methods section. Additionally, we have included a discussion of the study's limitations, including sample size, and their potential impact on the findings in the revised

We have also thoroughly reviewed and updated the reference list to ensure all citations are accurate and complete. All references are appropriately active and none of the papers has been retracted.

Thank you

Sincerely

Dr Kasagga Brian

---

## [Decision Letter · Decision Letter 2]

10 Jan 2025

Exploring Burn First Aid Knowledge and Water Lavage Practices in Uganda: A Cross-Sectional Study.

PONE-D-24-18839R2

Dear Dr. Kasagga,

We’re pleased to inform you that your manuscript has been judged scientifically suitable for publication and will be formally accepted for publication once it meets all outstanding technical requirements.

Kind regards,

Xiaosheng Tan

Academic Editor

PLOS ONE

Additional Editor Comments (optional):

Reviewers' comments:

Reviewer's Responses to Questions

**Comments to the Author**

1. If the authors have adequately addressed your comments raised in a previous round of review and you feel that this manuscript is now acceptable for publication, you may indicate that here to bypass the “Comments to the Author” section, enter your conflict of interest statement in the “Confidential to Editor” section, and submit your "Accept" recommendation.

Reviewer #1: All comments have been addressed

2. Is the manuscript technically sound, and do the data support the conclusions?

Reviewer #1: Partly

3. Has the statistical analysis been performed appropriately and rigorously? 

Reviewer #1: No

4. Have the authors made all data underlying the findings in their manuscript fully available?

Reviewer #1: No

5. Is the manuscript presented in an intelligible fashion and written in standard English?

Reviewer #1: Yes

6. Review Comments to the Author

Reviewer #1: I have no further commnets on this research paper.

7. PLOS authors have the option to publish the peer review history of their article (what does this mean?). If published, this will include your full peer review and any attached files.

Reviewer #1: No

---

## [Editor Report · Acceptance letter]

17 Jan 2025

PONE-D-24-18839R2 

PLOS ONE

Dear Dr. Kasagga, 

I'm pleased to inform you that your manuscript has been deemed suitable for publication in PLOS ONE. Congratulations! Your manuscript is now being handed over to our production team.

Kind regards, 

on behalf of

Dr. Xiaosheng Tan 

Academic Editor

PLOS ONE